# Predicting molecular subtypes of breast cancer using multimodal deep learning and incorporation of the attention mechanism

**Tianyu Zhang**[1,2]                                                              T.ZHANG@NKI.NL
**Luyi Han**[1,3]                                                                   L.HAN@NKI.NL
**Yuan Gao**[1,2]                                                                   Y.GAO@NKI.NL
**Xin Wang**[1,2]                                                                   X.WANG@NKI.NL
**Regina Beets-Tan**[1,2]                                                       R.BEETSTAN@NKI.NL
**Ritse Mann**[1,3]                                                                 R.MANN@NKI.NL

[1] *Department of Radiology, Netherlands Cancer Institute (NKI), Amsterdam, The Netherlands*

[2] *GROW School, Maastricht University, Maastricht, The Netherlands*

[3] *Department of Diagnostic Imaging, Radboud University Medical Center, Nijmegen, The Netherlands*

## Abstract

Accurately determining the molecular subtypes of breast cancer is an important factor for the prognosis of breast cancer patients, and can guide treatment selection. In this study, we report a multimodal deep learning with attention mechanism (MDLA) for predicting the molecular subtypes of breast cancer from mammography and ultrasound images. Incorporation of the attention mechanism improved diagnostic performance for predicting 4-class molecular subtypes with Matthews correlation coefficient (MCC) of 0.794. The MDLA can also discriminate between Luminal disease and non-Luminal disease with areas under the receiver operating characteristic curve (AUC) of 0.855. This work thus provides a noninvasive imaging biomarker to predict the molecular subtypes of breast cancer.

**Keywords:** Multimodal Deep Learning, Breast Cancer, Molecular Subtypes, Convolutional Neural Network, Attention Mechanism, Mammography, Ultrasound

## 1. Introduction

Breast cancer is the most common malignant tumor in women. It has a high degree of heterogeneity in terms of clinicopathological characteristics, prognosis, and response to treatment (Marchiò et al., 2020). Breast cancer can be divided into four molecular subtypes, which is usually based upon the expression levels of the estrogen receptor (ER), progesterone receptor (PR), human epidermal growth factor receptor 2 (HER2) and Ki-67. The resulting molecular subtypes are known as Luminal A, Luminal B, HER2+ and Triple-negative breast cancer (TNBC) (Goldhirsch et al., 2013). These molecular subtypes are important prognostic factor and can guide pre- and postoperative systemic therapy, because these typically target these receptors (Korde et al., 2021). At present, the molecular subtypes is determined by immunohistochemical indicators obtained from biopsy specimen (Ronchi et al., 2020), as a surrogate for genetic testing, as the genetic analysis is quite costly. Unfortunately, the biopsy limits the assessment to a small part of the tumor, which might prevent obtaining a full impression of the nature of the lesion (Jiang et al., 2020). Deep learning-based methods have been widely used in the field of breast imaging mainly for the evaluation of screening mammograms. Some studies have shown that deep learning-based methods can potentially be used to predict molecular subtypes of breast cancer (Jiang

et al., 2020). However, most previous studies only analyzed single-modality images and did not integrate features from different imaging modalities. In this work, we developed a deep learning-based model to predict molecular subtypes of breast cancer using multimodal image analysis, combining mammography (MG) and ultrasound (US). We show that the multimodal deep learning can potentially be used to predict the molecular subtypes of breast cancer.

## 2. Materials and Method

864 paired images studies (MG and corresponding US) of women with breast cancer presenting at the Netherlands Cancer Institute between January 2014 and November 2017 were retrospectively collected. The cancers were subdivided into 4 molecular subtypes based on the information of ER, PR, HER2 and Ki-67 from immunohistochemical findings, and included 382 cases of Luminal A, 272 cases of Luminal B, 74 cases of HER2+, and 136 cases of TNBC. All cases were randomly divided into training set and test set with a ratio of 4:1. Then, the multimodal deep learning algorithm based ResNet50 was developed to predict the molecular subtypes of breast cancer. This model was combined with the attention mechanism based on the work of Woo et al. (Woo et al., 2018) to create the final model (multimodal deep learning with attention mechanism: MDLA) (Figure 1).

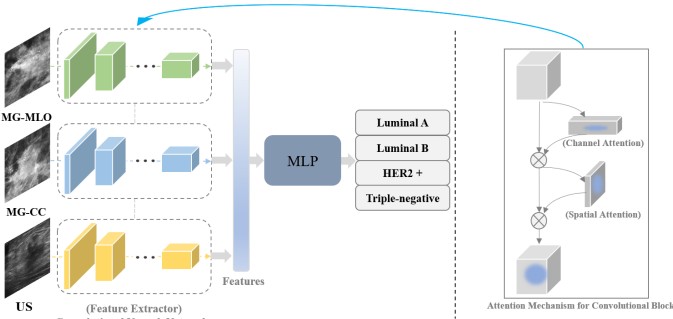

Figure 1: The scheme for predicting molecular subtypes of breast cancer.

Two views of MG (medio-lateral oblique -MLO- and cranio-caudal -CC-) and an US image of each case (lesion location) were used as input. All images were resized to $256 \times 256$ pixels and normalized before model training. In the training stage, data augmentation was applied to avoid overfitting, the Adam optimizer was applied to update parameters. We tested two classifications in this research: (1) To predict each of the 4 molecular subtypes of breast cancer, (2) To distinguish between Luminal (A and B) and non-Luminal disease (HER2+ and TNBC). Matthews correlation coefficient (MCC, ranging from -1 to 1, larger value means better performance) was used as an evaluation indicator for predicting 4-class molecular subtypes, and area under the receiver operating characteristic curve (AUC) was used to evaluate the performance of distinguishing between Luminal and non-Luminal.

## 3. Results

In the test set, the MCC was 0.681 and 0.729 for 4-class classifications using only MG or US images as input, respectively. Multimodal classification of breast cancer subtypes led to significantly higher accuracy. The MDLA model showed the best diagnostic performance

for predicting 4-class molecular subtypes with a MCC of 0.794, compared to a MCC of 0.765 for a model without integration of the attention mechanism. The confusion matrix for predicting 4-class molecular subtypes of breast cancer using MDLA is shown as Figure 2 a. The AUC was 0.855 [0.823, 0.887] for distinguishing between Luminal and non-Luminal using MDLA (Figure 2 b), compared to 0.838 [0.808, 0.868] for a model without the use of the attention mechanism.

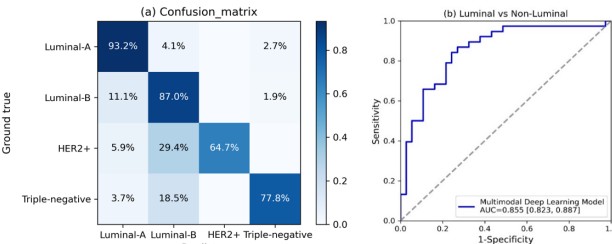

Figure 2: The confusion matrix and ROC curve for predicting molecular subtypes.

## 4. Discussion

In this study we developed the MDLA model, which can potentially be used to predict the molecular subtypes from combined mammography and ultrasound data. Multimodal imaging shows better performance than single-modal imaging, and the attention mechanism was shown to further improve the performance of our multimodal deep learning model. This supports the idea that combining multimodal medical imaging may provide more features to be used as image biomarkers for predicting molecular subtypes of breast cancer.

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
