# OpenReview forum: "Predicting molecular subtypes of breast cancer using multimodal deep learning and incorporation of the attention mechanism"
_MIDL.io/2021/Conference/Short — MIDL 2021 Poster_

### Official Review · Reviewer_p6QX · 2021-04-20

**Confidence:** 5
**Final Rating:** 4

**Summary:**

The authors propose to fuse a multi-path CNN with attention mechanisms to classify breast cancer images seen in MG and US into molecular subtypes. They train and test this approach on a dataset of multi-modal breast images and show how the attention blocks improve results over the baseline CNNs. The results of this experiment look promising by numbers.

**Strengths:**

Design and presentation of the work is very good. The paper is well-written and mostly clear. The clinical motivation is clear. The experimental setup is sound and clear.
A sufficiently large dataset has been used.

**Weaknesses:**

In particular when new methodological approaches are presented, I find it necessary to at least briefly highlight how the proposed new component (here: the attention mechanism) is fused into the existing backbone (here: the convolutional blocks). It should be explained which previous shortcoming is tackled with this, and ideally shown that the addition is sensible (e.g. by visually proving that the attention maps in the conv. blocks mask/highlight/suggest relevant areas for the subsequent conv. blocks).
As it is, I can only hypothesize about the motivation and benefit, and while it seems to be proven by the numbers, this alone does not convince me altogether (and does not give the authors any hints on how to improve, extend, modify their approach to optimize it!). Perhaps reasons for the frequent misclassification of HER2+ as luminal B could be explained through this.
Also, it would have been nice to mention the performance of previous approaches for image-based subtype classification (the authors cite one work).
Lastly, I would like to learn more about the MG resampling to the 256x256 size; assuming that the bounding box around lesions is adaptively sized to accomodate the lesion size, I would think that useful information might easily be lost in this step.

**Deanonymize Review:**

no

**Justification Of The Rating:**

This is in my view very solid work, based on a substantial dataset, a methodologically challenging idea to combine MG and US, a partially novel approach, and interesting results. I would like to see this presented and discussed.

**Paper Type:**

both

**Special Issue:**

no

---

### Official Review · Reviewer_P5aC · 2021-04-21

**Confidence:** 5
**Final Rating:** 2

**Summary:**

The main contribution of this work is a deep learning approach to classify breast cancer molecular subtypes.

Additional technical contributions include combining mammogram with ultra-sound images with well-known convolutional block attention modules used for  four-way classification of breast cancer subtypes.

**Strengths:**

Combining multiple views of mammogram and US both of which are routinely available during breast cancer screening are non-invasive methods for classifying breast cancer subtypes.

The results seem to be promising, although showing how this accuracy compares to some existing methods would be useful to see how the method performs.

A large number of patients were used in the analysis.




**Weaknesses:**

The methods are not novel. ResNet is a very known network and CBAM has been around since 2016.

Some details of the experimental setup could be presented better.

Comparison to other methods would be helpful to understand how this method fares.

Ablation tests are not shown to demonstrate how adding the attention module helps.

**Deanonymize Review:**

no

**Detailed Comments:**

The methods are not novel. ResNet is a very known network and CBAM has been around since 2016.

Some details of the experimental setup could be presented better. For example, how was the evaluation done. Was set aside testing used to report the accuracy? Also in general, the distribution of the different molecular sub-types is very unbalanced. So reporting per class accuracy is important. Only the AUC metric is shown which is not the best metric to show how the classification was for each sub-type.

Comparison to other methods would be helpful to understand how this method fares. Dr. Madabhushi's group, Dr. Sutton have all published papers on this topic, using non-deep learning and deep learning? methods. Would be good to at least discuss the accuracy comparisons.

Also, ablation tests are missing - this is minor because this is a short paper. Nevertheless, it is important to understand if the attention module was indeed useful.

**Justification Of The Rating:**

The method is not novel. It uses existing methods like ResNet and CBAM.

More details of the experiment is necessary. Also additional metrics should be shown to demonstrate that the classifier is unbiased to the majority class alone.

**Paper Type:**

validation/application paper

**Special Issue:**

no

---

### Meta-Review · Program_Chairs · 2021-05-11

**Recommendation:** Accept (Poster)
**Confidence:** 5

**Metareview:**

This paper is a decent paper and mostly clear. The authors are suggested to address all the review comments in final version.

---

### Decision · Program_Chairs · 2021-05-11

Accept (Poster)